# “I Feel like I’m Eating Rice 24 Hours a Day, 7 Days a Week”: Dietary Diversity among Asylum Seekers Living in Norway

**DOI:** 10.3390/nu11102293

**Published:** 2019-09-26

**Authors:** Sigrun Henjum, Bess L. Caswell, Laura Terragni

**Affiliations:** 1Department of Nursing and Health Promotion, Faculty of Health Sciences, OsloMet—Oslo Metropolitan University, 0130 Oslo, Norway; 2Department of Nutrition, University of California, Davis, CA 95616, USA; blcaswell@ucdavis.edu

**Keywords:** Dietary diversity, food security, asylum seekers, hunger, Norway

## Abstract

Food insecurity is widespread among asylum seekers resettled in Western countries. Limited information exists on the quality of food intake in this population. The aim of this study was to investigate dietary quality among asylum seekers living in Norwegian reception centers. This study has a cross-sectional research design. Dietary intake was assessed through a qualitative 24-hour dietary recall, and the dietary diversity score (DDS) was calculated. This study was conducted in eight Norwegian reception centers. A total of 205 adult asylum seekers (131 men and 74 women) participated in the study. The asylum seekers ate on average two meals per day, and one-third ate their first meal after noon. Mean (SD) DDS was 4.0 (1.6) and 2/3 had low dietary diversity, eating from fewer than five food groups. Women had a significantly higher mean DDS (4.5) than men (3.8) (β (95% CI): 0.47 (0.00, 0.95) and a higher consumption of vegetables and fruits. The longer the period of residence in Norway, the higher the DDS, β (95% CI): 0.01 (0.00, 0.02). The asylum seekers’ inadequate dietary intake reveals new forms of poverty and social exclusion in Europe. An inadequate dietary intake may increase the magnitude of difficulty involved in the settlement process and contribute to poorer health.

## 1. Introduction

Poor nutrition among immigrants, refugees, and asylum seekers living in Western countries is widespread and regarded as a serious problem [1,2,3,4]. In particular, inadequate dietary intake may have serious implications on health, such as impaired work capacity, nutrient deficiency diseases, and an increased risk of pregnancy-related complications in women [5,6,7]. In addition, asylum seekers are more likely to experience the double burden of malnutrition, due to preexisting micronutrient deficiencies, as well as a high risk for chronic diseases due to higher intakes of sodium, fat, and sugar in the new country [8]. Lack of money, unfamiliarity with new foods and new language are among the main challenges that asylum seekers experience upon resettlement [9,10,11,12]. Existing studies on the health status of refugees and asylum seekers in Europe have stated that they are at risk of suffering a decline in health because of poor housing arrangements and inadequate dietary intakes [4,13,14]. 

Most studies investigating the dietary intakes of asylum seekers in Western countries have focused on food security. According to the World Health Organization, food security can be defined as “access to food of sufficient quantity and quality for all household members at all times through socially acceptable means” [15]. Food insecurity is a key indicator for evaluating contextual aspects related to food consumption (e.g., accessibility to nutritious food, uncertainty about food provision, etc.) and providing indications of malnutrition and hunger [16]. However, measures of food security provide little information on dietary intake and dietary quality. Other approaches are needed to create a more complete picture of individuals’ dietary intakes. Dietary diversity score (DDS) is a qualitative measure of food consumption counting the number of food groups consumed, reflecting household access to a variety of foods, and serving as a proxy for nutrient adequacy of the dietary intake [17]. The consumption of a higher number of food items and food groups is associated with improved nutritional adequacy [18,19,20] and better health [7].

In the last ten years, the number of asylum seekers heading to Europe has dramatically increased [21]. In Norway, at the start of the study in January 2017, 12,674 asylum seekers were residing in Norwegian reception centers [22]. While waiting for their application to be processed, asylum seekers live in reception centers; they are not allowed to work but receive an allowance from Norwegian authorities [23,24]. Most ordinary reception centers have self-catering, which means that the residents themselves do the grocery shopping and meal preparation. Guidelines from the Ministry of Health in Norway state that reception centers should give residents information about a healthy diet and consumption of foods that meet basic nutritional needs [22]. Norway is also committed, through the United Nations International Covenant on Economic, Social and Cultural Rights, to ensure that the right to adequate food and health are respected. Despite these commitments, previous studies conducted in Norway have provided indications of a precarious nutrition situation, like poor kitchen facilities, a limited amount of money to buy food, distance from shops and capability of preparing nutritious meals among residents in reception centers [25,26]. An explorative qualitative study conducted previously [26] showed how food can be become part of a “politic of inhospitality”. The experiences with food in asylum reception centers reinforced feelings of precariousness and made asylum seekers not feel at home or welcome [26]. Studies on food in the context of migration have most often focused on women. This is understandable given the role that food work has in defining women’s identities as caregivers, particularly in traditional societies [27]. However, it is important to point out that in the case of asylum seekers, it is more likely for men to travel alone [22]. Yet, little is known about how men cope with food in an unfamiliar environment and about their actual diets. 

To the best of our knowledge, there are currently no studies describing dietary intake and evaluating dietary diversity and food practices among asylum seekers in Norway. Therefore, the aim of this study was to address this gap in the literature. We also wanted to assess possible gender differences in DDS and food practices. More knowledge on this topic can contribute to the design of programs to improve dietary intake and health in this vulnerable group.

## 2. Materials and Methods 

### Sample Selection 

We contacted nine ordinary asylum reception centers in the southeastern part of Norway and sent out invitation letters to participate in the study. Asylum reception centers in southeastern Norway were chosen since they were within travel distance from Oslo by train or bus. In order to have variation in accessibility to food, we contacted reception centers located in urban areas and in rural areas. A reception center was considered rural if the nearest grocery store was more than four kilometers away, which is far higher than the commonly used definition of a walking distance of 450 meters [28]. Eight reception centers (four urban and four rural) responded positively and were included in the study. Participants in the survey were recruited through convenience sampling. The aim was to include asylum seekers from the most represented countries and participants of both genders. Adult men and women (at least 18 years old) were asked to join the study. Among families with more than one adult, the first adult to volunteer was selected to participate. Fieldworkers speaking the languages of the participants (i.e., Arabic, Somali, Dari, and Tigrinja, in addition to Norwegian and English) were employed for the project. Fieldworkers recruited participants through the information meetings at the reception centers (lobbies, information office, kitchens). Data were collected from 31 January to 25 August 2017; however, data collection was suspended during Ramadan (June), because dietary habits would probably change during fasting. 

The questionnaire consisted of four parts: a 12-question module on socioeconomic status, a 24-hour dietary recall, a 19-question module on food skills and a 10-question food security assessment. Food security was measured by the 10-item Radimer/Cornell Hunger and Food Insecurity Scale, a tool which captures household-, individual- and child-level experiences of hunger or food insecurity [29]. The scale consists of 10 questions dived into three groups: food insecurity without hunger (4 questions), food insecurity with hunger (4 questions), and food insecurity with child hunger (2 questions). Each question had three possible answers: “never”, “sometimes”, and “often”. Participants were categorized as food-insecure if they responded “sometimes” or “often” to at least one question. The method used to measure food insecurity in this population has been described in detail in a previous publication [2]. To collect data on income, we asked every participant to self-report their total monthly budget. We asked questions about food practices like number of meals per day and the timing of the first meal. The questionnaire was administered by field workers who received training in interview techniques from the project managers. The field workers translated the questionnaire from English into Arabic, Somali, and Dari. The project managers developed the questionnaire based on already validated questionnaires, but some changes were made after pilot testing and discussions with field workers to adapt it to the conditions of asylum seekers. Changes were made particularly in the food skills section. For instance, in the question “How often do you buy food”, the option “When I got the allowance from the public authorities” was added to the list of already existing answers (i.e., every day, every week). A question about the availability of familiar food in the grocery shop was also added. In this paper, the data presented are primarily from the qualitative 24-hour recall.

Dietary intake was assessed by a single qualitative 24-hour dietary recall. During the recall, participants were asked to report all the foods and beverages consumed the preceding day, as well as the time of consumption. Names of mixed dishes, cooking method and ingredients used were recorded. Amounts consumed were not reported. A dietary diversity score (DDS) was calculated to assess dietary quality. The score is defined as the number of food groups consumed during the recall day. Mixed dishes were disaggregated into ingredients before calculating the DDS. The present paper used the food groups from Minimum Dietary Diversity—Women (MDD–W) introduced in 2014 by the Food and Agriculture Organization (FAO) of the UN and the Food and Nutrition Technical Assistance Project (FANTA) [17,30]. The MDD–W uses ten food groups: starchy staple foods; beans and peas; nuts and seeds; dairy; flesh foods; eggs; vitamin A-rich dark green leafy vegetables; other vitamin A-rich vegetables and fruits; other vegetables; and, other fruits. Consumption of five or more food groups out of ten is considered the minimum dietary diversity, indicating a greater likelihood of meeting micronutrient needs [30]. We measured intake of three additional food groups: salty snacks, sweets and sugar drinks. The number of meals and the time of consumption of the first meal were recorded. The first meal was defined as the first food item or drink consumed, meaning that if a person drank a cup of tea, it was registered as the first meal. 

The Statistical Package for the Social Sciences version 23.0 (IBM Corp., Armonk, New York, NY USA) was employed to analyze the data. P-values lower than 0.05 were considered significant. Normally distributed continuous data were presented as means and standard deviations. To test for differences by group in continuous variables, independent sample t-tests were used. Chi-square tests were used for categorical variables. Linear regression analyses were performed to assess associations between DDS (outcome variable) and the selected socioeconomic variables (exposure variables). Potential exposure variables were purposefully selected in advance based on theoretical mechanisms of influence on dietary practices [31]. The potential exposure variables were age, gender, number of children, location of reception center (urban/rural), status of application (accepted vs. rejected/other), self-reported monthly budget (continuous variable in euros, including both the received allowance for each family member and income from other sources) and months of stay in Norway. All the covariates showing a linear association (*p* < 0.10) in the crude regression models were included in a preliminary multiple regression model. Those that were still significantly associated in this model (*p* < 0.10) were retained in the final model. The regression models were checked for homoscedasticity using standard residuals within ±3 and cook’s distance <1 as parameters.

## 3. Results

In total, 205 asylum seekers (26% of the asylum seekers registered as residents in the visited reception centers) participated in the study. Mean age was 31 years and 36% were female (Table 1). The participants came from Syria (25%), Eritrea (18%), Somalia (11%), Iraq (11%) and others. The mean length of stay in Norway was 29 (range 1–168) months, with 40% having had their application to remain granted and 30% having had their application rejected. Only 11% of the men lived with children, compared to 41% of the women. 

The mean monthly budget for men and women was 243 and 275 euro, respectively, when dividing monthly budget on number of children. Ninety-three percent of the asylum seekers were categorized as food insecure (95% of men and 88% of women).

The mean DDS among the asylum seekers was 4.0 (±1.6), and women had a significantly higher DDS than men, 4.5 (±1.7) and 3.8 (±1.5), respectively (Table 2). The number of food groups consumed ranged from one to nine for women and from zero to seven for men. Sixty percent of the asylum seekers had a DDS below the minimum cutoff of five food groups (66% of men and 49% of women).

The most commonly consumed food groups were starchy staple foods, vegetables, flesh foods, and dairy. Consumption of sugary drinks was also common. Vitamin A-rich dark green leafy vegetables, beans/peas, nuts/seeds, and eggs were the least consumed food groups (Table 3). Significantly more women than men consumed vitamin A-rich dark leafy vegetables, other vitamin A-rich vegetables and fruits, and other fruits. Compared to food secure asylum seekers, a smaller percentage of food insecure asylum seekers consumed milk and dairy products, other vegetables and salty snacks. The mean number of meals among the asylum seekers was 2.3 ± 0.8, with no significant differences between men and women. Thirty-five percent of the asylum seekers ate their first meal after noon. 

In order to understand the factors influencing dietary intake, a multiple linear regression model was used to identify determinants of DDS (Table 4). In the final model, including age, gender and months of stay in Norway, female gender and length of stay in Norway were positively associated with DDS, 0.47 (0.00, 0.95) and 0.01 (0.00, 0.02), respectively. 

## 4. Discussion

Our study shows that two-thirds of the asylum seekers had a low dietary diversity as they were eating from fewer than five food groups. The low dietary diversity in this population is comparable to dietary intakes found in low-income countries, such as Nepal [32] and Bangladesh [33], as well as among refugees in Algeria [34]. The limited dietary intake among asylum seekers is in sharp contrast to the rest of the Norwegian population who, largely, have access to abundant food [35,36]. Food insecurity and limited dietary intake are a well-known problem throughout the world, and its magnitude in low-income countries is highly visible [37]. In high-income countries, its presence is often less noticeable. However, recent studies have pointed out that food insecurity is a rising problem in affluent societies and that refugees are among the most vulnerable groups [38,39]. 

Prior research has documented that vulnerable households reduce their food expenditures by limiting or excluding specific food items from their diet [40,41,42,43]. This tendency was visible in our study; the asylum seekers consumed few food items and had a monotonous diet. Similar findings have been reported in Sweden [44], Australia [45,46], the UK [13], and the US [1,3,47,48]. Somali women in Melbourne had diets which did not provide the recommended intakes for multiple nutrients [49]. In Norway, a study on single minor asylum seekers found that they often prioritized cheap food items with low nutritional value [50]. Another study on asylum seekers in Sweden revealed that they mainly bought staple foods and many could not fulfill their nutritional needs [44]. Limited dietary intake and a monotonous diet pose long- and short-term health risks for asylum seekers [5,6].

Our study also indicates that asylum seekers living in reception centers tended to eat few meals and many did not consume their first meals before noon. Eating fewer meals has emerged as one of the strategies among families suffering from financial setbacks [40,41]. In addition, the living conditions in the asylum reception centers may have an influence on meal structure [26,51]. Several studies have indicated that life at asylum reception centers is characterized by a lack of activities, loneliness, and social isolation [52,53]. Gathering for a meal is an opportunity for socialization in everyday life, but for asylum seekers, the social aspect of meals often disappears, as well as the motivation to cook [26,44]. This can lead to apathy and depression and affect appetite. The housing standard at reception centers and the limited kitchen facilities could also contribute to reducing meal preparation [25]. The kitchen facilities at the reception centers in our study had limited storage space, poor hygiene, and were crowded, with several families having to share a kitchen. 

The findings from our study reveal that among food-insecure asylum seekers, intakes of milk, vegetables and salty snacks were significantly lower than among food-secure households, which was observed among Somali refugees in the US [54]. This may be due to the lower cost of calorie-dense foods high in fat and sugar compared to nutrient-rich foods [55]. Sodas and snacks are often considered “luxury” items in low-income countries; however, easier access and the relatively low cost of these foods in high-income countries may allow food-secure households to afford them [3]. In addition to the cost, the notion that vegetables and fruits do not satisfy hunger may keep low-income households from buying them. In some cultures and countries, such as Somalia, meat is regarded as the main part of a meal. When faced with financial constraints, Somali families may prioritize meat over fruits and vegetables [3]. 

In our study, asylum seekers’ meals tended to consist of food that was known in their country of origin and there were few indications of consumption of Norwegian food such as for instance pollock, roots vegetables, cabbage. For those who are far away from their homeland and often separated from their relatives, maintaining familiar food habits can be a way to rebuild a sense of home and cope with many of the problems faced in a new country [27,56,57]. However, the maintenance of a traditional diet and the consequent limited food items used for preparing meals can also be due to a lack of knowledge about the food available in Norwegian shops or fear of eating foods prohibited by their religion [10,57]. 

We found that gender and length of stay in Norway were associated with DDS among the asylum seekers in our study. DDS was significantly higher among women than among men, with a higher consumption of vegetables and fruits. A higher consumption of fruits and vegetables among women is also observed in the general Norwegian population, reflecting women’s more positive attitudes toward healthy eating [58]. In addition, women often have better cooking skills than men, and such skills are correlated with improved dietary intakes [59,60]. Men coming from countries characterized by traditional gender roles and who are living alone in reception centers may struggle more than women in preparing meals [10,27]. Although in general, women have better cooking skills, they still may encounter difficulties in adapting these skills in the new situation because of the limited kitchen facilities at the centres and limited knowledge of food purchasing practices in the new context [10,12,45,61]. Among refugees who had lived in Australia for less than 12 months, lack of budgeting and household skills were among the causes of high prevalence of food insufficiency [45]. 

The mean length of stay in Norwegian asylum reception centers was 29 months, and asylum seekers who had stayed longer in Norway had a significantly higher DDS than their counterparts. In Norway, asylum seekers have the right to live in an ordinary reception center while their application is processed. Many asylum seekers live in such centers for a long period, some even for years [22]. Length of stay may contribute to better familiarity with the new food environment in the host country. 

Difficult economic situations may force asylum seekers to buy the cheapest foods, items that are on sale or expired, and quantity over quality. Studies have that shown socio-economic status is positively associated with dietary diversity [34,61]. However, after adjusting for possible confounders, there was no difference in income between asylum seekers with a high and low DDS in our study. In Norway, an asylum seeker receives approximately 250 euros per month, and this sum is supposed to cover all expenses other than accommodation, including food, clothing, transport and any other necessities [24]. In contrast, the Norwegian reference budget recommends approximately 250 euros to cover one adult’s food costs for one month, and an average Norwegian family uses 11% of its budget on food [62]. One explanation for the lack of association between income and DDS could be the fact that all the asylum seekers were low-income families or individuals according to Norwegian standards; it was not therefore possible to differentiate between groups with higher and lower incomes. 

### Strengths and Limitations

This study provides novel information on dietary intake and quality among asylum seekers in Norway. Strengths of the study include the high number of participants compared with similar studies and the inclusion of men and women from different countries of origin. Data quality and participation were supported by field workers who could conduct recruitment and interviews in languages spoken in the asylum seekers’ countries of origin. 

The study has also some limitations. First, the convenience sampling strategy used to recruit participants prevents generalization of the findings to all asylum seekers in Norway and introduces the possibility of a selection bias. However, the gender distribution in our study represented the gender distribution of asylum seekers in Norway in 2017 [63] and participants from all the main countries of origin were included. In addition, the findings in our study are supported by the findings of a previous qualitative study [26] and may therefore be indicative of the conditions experienced by other asylum seekers. Secondly, the qualitative 24-hour recall technique used to collect dietary data is only reflective of a single day’s intake, cannot be used to estimate quantities of foods or nutrients consumed and is dependent on participant memory and communication skills. With qualitative recall data, diet quality indices more commonly used in higher-income countries, such as the Healthy Eating Index [64], cannot be calculated. We found few recent studies using dietary diversity scores describing dietary patterns among migrants or refugees from low- or middle-income countries [65,66,67]. However, the advantages of the qualitative recall for the purposes of this study were that it relies only on short-term memory, had low participant burden, and can be used among groups with low literacy or numeracy. In our study, the data show a range of DDS from 1 to 7, which supports our use of this score to capture variation in dietary quality within this nutritionally vulnerable population.

## 5. Conclusions

To the best of our knowledge, this is the first study on dietary intake and quality among asylum seekers in Norway, and it is one of the first in Europe. By assessing their dietary intake, we revealed that asylum seekers had a monotonous diet with low variety and few meals. This is in contrast to the food abundance to which most Norwegians are accustomed, revealing, as other studies have indicated, the emergence of new vulnerable groups and new forms of poverty and exclusion in high-income countries with otherwise strong welfare systems. The situation is particularly critical given Norway’s commitment to ensuring the right to adequate food and health, as outlined in the UN International Covenant on Economic, Social and Cultural Rights.

For asylum seekers already struggling with social and familial disruptions, an inadequate dietary intake may increase the difficulty involved in the resettlement process and contribute to poorer health. Therefore, initiatives aimed at ameliorating the food situation at asylum reception centers are needed. Improving food security could promote the maintenance of cultural identity, facilitate the transition into a new food environment, and could support both short- and long-term health [43]. These measures should address both the structural barriers related to poor economic and living conditions, as well as improve knowledge about foods available in Norwegian stores and cooking skills. 

## Figures and Tables

**Table 1 nutrients-11-02293-t001:** Background information on asylum seekers in reception centers in southeastern Norway (*n* = 205) by gender and in total.

	Men (*n* = 131) *n* (%) or Mean ± SD	Women (*n* = 74) *n* (%) or Mean ± SD	Total (*n* = 205) *n* (%) or Mean ± SD
**Age (years)**	29.7 ± 9.0 *	33.3 ± 11.7 *	31 ± 10.2
**Country of origin**			
Eritrea	17 (13.0) *	19 (25.7) *	36 (17.6)
Iraq	16 (12.2)	6 (8.1)	22 (10.7)
Somalia	9 (6.9)	13 (17.6)	22 (10.7)
Syria	45 (34.4) *	7 (9.5) *	52 (25.4)
Other	43 (32.8)	29 (39.2)	72 (35.1)
**Marital status**			
Married	42 (32.1) *	37 (50.0) *	79 (38.5)
Single	77 (58.8) *	18 (24.3) *	95 (46.3)
Other	12 (9.2)	19 (25.7)	31 (15.1)
**Education completed**			
None	16 (12.2)	23 (31.1)	39 (19.0)
1–12 years	64 (48.9)	30 (40.5)	94 (45.9)
Higher	51 (38.9)	20 (27.0)	71 (34.6)
**Location of reception center**			
Urban	79 (60.3)	41 (55.4)	120 (58.5)
Rural	52 (39.7)	33 (44.6)	85 (41.5)
**Living situation**			
Alone	28 (21.4)	12 (16.2)	40 (19.5)
Husband/wife without children	3 (2.3)	2 (2.7)	5 (2.4)
With children	14 (10.7) *	30 (40.5) *	44 (21.5)
With other not family	86 (65.6) *	30 (40.5) *	116 (56.6)
**Asylum application status**			
Submitted	41 (31.3)	22 (29.7)	63 (30.7)
Accepted	56 (42.7)	25 (33.8)	81 (39.5)
Rejected	34 (26.0)	27 (36.5)	61 (29.8)
**Monthly budget euro** **Months of stay in Norway**	292 ± 174 * 23.1 ± 24.3 *	369 ± 216 * 39.0 ± 38.3 *	320 ± 193 28.8 ± 30.9
**Living with children**	14(10.7) *	30 (40.5) *	44 (21.5)
**Food insecure**	125 (95.4) *	65 (87.8) *	190 (92.7)

* *p* < 0.05 for chi-square test (categorical variables) or t-test (continuous variables) for difference between men and women.

**Table 2 nutrients-11-02293-t002:** Distribution of dietary diversity scores among men and women asylum seekers living in reception centers in southeastern Norway (*n* = 205).

Dietary Diversity Score	Men (*n* = 131) *n* (%)	Women (*n* = 74) *n* (%)	Total (*n* = 205) *n* (%)
0	2 (1.5)	0 (0.0)	2 (1.0)
1	5 (3.8)	3 (4.1)	8 (3.9)
2	18 (13.7)	8 (10.8)	26 (12.7)
3	33 (25.2)	11 (14.9)	44 (21.5)
4	29 (22.1)	14 (18.9)	43 (21.0)
5	27 (20.6)	18 (24.3)	45 (22.0)
6	13 (9.9)	11 (14.9)	24 (11.7)
7	4 (3.1)	7 (9.5)	11 (5.4)
8	0 (0.0)	1 (1.4)	1 (0.5)
9	0 (0.0)	1 (1.4)	1 (0.5)

**Table 3 nutrients-11-02293-t003:** Frequency of food group consumption in the past 24 h among asylum seekers living in reception centers in southeastern Norway by gender and food security status (*n* = 205).

Food Groups	Men (*n* = 131) *n* (%)	Women (*n* = 74) *n* (%)	Food Secure (*n* = 15) *n* (%)	Food Insecure (*n* = 190) *n* (%)	Total (*n* = 205) *n* (%)
**10 food groups**					
All starchy staple foods	124 (94.7)	72 (97.3)	14 (93.3)	182 (95.8)	196 (95.6)
Beans and peas	41 (31.3)	19 (25.7)	4 (26.7)	56 (29.5)	60 (29.3)
Nuts and seeds	5 (3.8)	3 (4.1)	1 (6.7)	7 (3.7)	8 (3.9)
Dairy	64 (48.9)	46 (62.2)	13 (86.7)	97 (51.1) *	110 (53.7)
Flesh foods	71 (54.2)	41 (55.4)	10 (66.7)	102 (53.7)	112 (54.6)
Eggs	44 (33.6)	16 (21.6)	3 (20.0)	57 (30.0)	60 (29.3)
Vitamin A-rich dark leafy vegetables	9 (6.9)	15 (20.3) *	2 (13.3)	22 (11.6)	24 (11.7)
Other vitamin A-rich vegetables and fruits	5 (3.8)	19 (25.7) *	3 (20.0)	21 (11.1)	24 (11.7)
Other vegetables	92 (70.2)	56 (75.7)	15 (100.0)	133 (70.0) *	148 (72.2)
Other fruits	42 (32.1)	43 (58.1) *	7 (46.7)	78 (41.1)	85 (41.5)
**Additional food groups**					
Salty snacks	9 (6.9)	2 (2.7)	3 (20.0)	8 (4.2) *	11 (5.4)
Sweets	38 (29.0)	23 (31.1)	3 (20.0)	58 (30.5)	61 (29.8)
Sugary drinks	85 (64.9)	53 (71.6)	12 (80.0)	126 (66.3)	138 (67.3)

* Differences tested with chi-square, *p* < 0.001.

**Table 4 nutrients-11-02293-t004:** Multiple linear regression model of dietary diversity score (DDS) by background characteristics among asylum seekers living at reception centers in southeastern Norway (*n* = 205).

Variables	Unadjusted Beta Coefficients ^a^ (95% CI)	*p*	Adjusted Beta Coefficients ^b^ (95% CI)	*p*
Age (years)	0.03 (0.01, 0.05)	0.02	0.01(−0.01, 0.04)	0.32
Gender (men vs. women)	0.67 (0.21, 1.12)	<0.001	0.47 (0.00, 0.95)	0.05
Length of stay (months)	0.01 (0.01, 0.02)	<0.001	0.01 (0.00, 0.02)	0.03
Monthly budget (euros)	0.13 (0.01, 0.25)	0.03	0.08 (−0.05, 0.21)	0.22
Number of children	0.19 (0.03, 0.35)	0.02	0.09 (−0.09, 0.28)	0.33

^a^ Univariate analyses included the following variables; age, gender, number of children, location of reception center (urban/rural), status of application (accepted vs. rejected/other), self-reported monthly budget and months of stay in Norway. ^b^ Adjusted for the following variables; age, gender, length of stay in Norway, monthly budget and number of children.

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
