# Peer review of "“I Feel like I’m Eating Rice 24 Hours a Day, 7 Days a Week”: Dietary Diversity among Asylum Seekers Living in Norway"

_nutrients, 2019, doi:10.3390/nu11102293_

Round 1

Reviewer 1 Report

Thank you for making the suggested changes, I think the revised article is scientifically sound and much clearer.

Line 58: Dietary diversity score is a qualitative measure, not dietary diversity in itself, please add “score” to your sentence

Line 88 : please define DDS as it has not been defined yet

Line 156: please replace “n” by “in”

Line 167: Added? The sentence is incomplete

Line 578: Please consider replacing “from their relatives to maintain familiar” by “from their relatives, maintaining familiar”

Line 592: “can result unfamiliar » this part of the sentence does not make sense, something is missing

Line 601: there should not be a coma between situations and may

Line 755: I would specify after adjusting for …, because it is significant in univariate analysis

Line 780: consider replacing “to describe” by “describing”

Author Response

Thank you for the positive feed backs to our revision and for the last comments

Reviewer 2 Report

This is an interesting study which contributes to the literature in this discipline. I am grateful to the authors for addressing my original feedback which makes for a stringer resubmission. I am similarly grateful for the explanatory note around each comment. The strengths and limitations of the paper are articulated exceptionally well.

Minor additional proofreading is required as below:

Line 79 - Delete "ing" (ie) "... them not feel at home..."

Line 105 - Replace 'was' with 'were' (ie) "... data were collected..."

Line 156 Should read as "...in this population..."

Line 180 - Should read as "were recorded"

Line 757 - Delete space between transport and comma (ie) "transport,"

Author Response

Thank you for the positive feed backs to our revision and for the last comments

This manuscript is a resubmission of an earlier submission. The following is a list of the peer review reports and author responses from that submission.

Round 1

Reviewer 1 Report

This is a really interesting article. It reveals an important problem among asylum seekers in an objective way. This is critical to improve their living conditions and prevent poor health.

Line 53: “there were 12,674 asylum seekers residing” or “12,674 asylum seekers were residing”

Line 77: please add an “s” to participant

Line 77-78: please reformulate this sentence.

Line 79-80: When there was more than one person per household, how did you choose the one who participated. Please discuss how this may have influence your results in the discussion

Line 75-76 and 82-83: this is similar information, please group these sentences

Line 88: I do not understand what the information in the parentheses means

Line 133-134: Including only variables that are significantly correlated in bivariate analysis increase the risk of overfitting your multivariate model. Please discuss or make appropriate modification to the statistical analyses performed (e.g. include variables in the multivariate model based on theoretical knowledge and keep then even if they are not significantly associated). What is described here does not match Table 5

Table 1: Dairy or milk and dairy?

Table 2: Food insecurity number is not a percentage and what does the b stand for?

Line 145-146: It would be more relevant to present the proportion of men and women who are food insecure rather than the proportion of men and women among those who are food insecure.

Line 150-151: Similar comment to the previous one

Table 3: Consider presenting this information in a bar-graph

Tables 2-3: what is presented exactly, n % Mean (SD) as written or n (%)? This is not clear

Line 163: Why was 11 AM chosen as a threshold?

Line 166: please change to chi-square

Table 5: How was country of origin included in the model? It should be done with multiple dummy variables. Application granted: what is the reference group. Please state the proportion of the variance that is accounted for by the models

Line 211: I did not see the proportion of asylum seekers who did not eat before noon. The proportion not eating before 11 AM is presented.

Line 220: please reformulate “and were crowded”

Line 222: was the food security questionnaire targeting household or the participant? All snacks or salty snacks, please specify

Line 265-272: Consider moving this paragraph to the conclusion

Strengths and limitations: To my knowledge, the DDS has been seldom used in higher income countries. Please discuss the adequacy of this proxy. Please also discuss the possible impact of fieldworkers on results, e.g. their gender, were they originally from similar countries, asylum seekers themselves, etc. The fact that they spoke different languages is a strength of the study.

General comment: There are many comments in the discussion that posit that asylum seekers do not have sufficient cooking skills. This generalization may apply to men who come from gendered-based countries but in these countries, women knew how to cook, they have been doing so for generations. I would specify that they may lack specific cooking skills that may be different in Norway (e.g. using a gas-stove instead of an open-pit) or lack of food literacy in the Norwegian context.

Reviewer 2 Report

This paper contributes to the published literature on what is known in the food insecurity discipline regarding consumption and dietary quality generally, and for particular vulnerable consumer sub-groups specifically (asylum seekers). It is an interesting and worthwhile read that makes important observations throughout.

I would suggest that the literature review includes additional context regarding the authors' secondary aim regarding gender differences in DDS and food practices to strengthen this aspect of the paper. Furthermore, the addition of published literature around asylum seekers' substitution of dietary custom foods by Norwegian/local foods would be welcome context. There is scope to introduce earlier reference 55 (line 235) to overcome this literature omission. 

Similarly, is there a rationale supported by the literature to explain the authors' choice of 4kms distance as the threshold for rurality?

Additional narrative around the adaptations made by the authors to the questionnaire would further strengthen the Materials and method section (line 101).

Are there comparable food insecurity figures for the general Norwegian population to serve as a useful comparator to the asylum seeker population's experience of this phenomenon?

Did the authors conduct any qualitative research as part of the wider study? Are the earlier / subsequent papers exploring the qualitative experience of asylum seekers' dietary practices? It would be a worthwhile addition to this paper to signpost the reader to any qualitative exploration  of asylum seekers' views on their dietary practices of consuming a monotonous diet of few food items. Additionally, (line 252) qualitative signposting/synopsis of research findings regarding how length of stay contributes to asylum seekers' familiarity with indigenous foods would further strengthen this paper.

The authors annotate well the limitations and strengths of their study.

Minor additional proofreading required, as below:

Line 53: Delete redundant use of "were" so that were residing" reads as "residing"

Line 60: Delete redundant use of "of" so that "of among residents" reads as "among residents"

Line 69: Delete extra period.

Line 76: Should read as "were held" 

Line 78: Should read as "most represented"

Line 86: Should read  as "no data were collected"

Line 88: Should this read as "not within walking distance"?

Line 95: Should read as  "We asked questions"

Line 96: Should read as "number of meals"

Line 116 Should re as "were recorded"

Line 172: Insert comma before 'bharat'

Line 294: Delete extra period.

Line 299: Should this read as Public Health Nutrition Research Group?